# Identification of Skin Lesions by Snapshot Hyperspectral Imaging

**DOI:** 10.3390/cancers16010217

**Published:** 2024-01-02

**Authors:** Hung-Yi Huang, Hong-Thai Nguyen, Teng-Li Lin, Penchun Saenprasarn, Ping-Hung Liu, Hsiang-Chen Wang

**Affiliations:** 1Department of Dermatology, Ditmanson Medical Foundation Chiayi Christian Hospital, Chia Yi City 60002, Taiwan; 07117@cych.org.tw; 2Department of Mechanical Engineering, National Chung Cheng University, 168, University Rd., Min Hsiung, Chia Yi City 62102, Taiwan; 3Department of Dermatology, Dalin Tzu Chi General Hospital, No. 2, Min-Sheng Rd., Dalin Town, Chia Yi City 62247, Taiwan; tanglilin1121@hotmail.com; 4School of Nursing, Shinawatra University, 99 Moo 10, Bangtoey, Samkhok, Pathum Thani 12160, Thailand; pencbun_sa@mru.ac.th; 5Division of General Surgery, Department of Surgery, Kaohsiung Armed Forces General Hospital, 2, Zhongzheng 1st.Rd., Lingya District, Kaohsiung City 80284, Taiwan; 6Director of Technology Development, Hitspectra Intelligent Technology Co., Ltd., Kaohsiung City 80661, Taiwan

**Keywords:** skin cancer detection, hyperspectral imaging, mycosis fungoides, psoriasis, atopic dermatitis

## Abstract

**Simple Summary:**

This research revolutionizes dermatological diagnostics by integrating artificial intelligence (AI) and hyperspectral imaging (HSI) to identify skin cancer lesions, particularly Mycosis fungoides (MF). It differentiates MF from conditions such as psoriasis and atopic dermatitis using a dataset of 1659 skin images. This study employs a novel AI algorithm alongside advanced techniques for precise lesion segmentation and classification, moving diagnosis from color to spectral analysis. This non-invasive and efficient method marks a significant advancement in the early and accurate detection of skin malignancies. The model’s high performance is validated by its sensitivity, specificity, and accuracy, making it a vital tool in dermatology for identifying skin cancers and inflammatory conditions.

**Abstract:**

This study pioneers the application of artificial intelligence (AI) and hyperspectral imaging (HSI) in the diagnosis of skin cancer lesions, particularly focusing on Mycosis fungoides (MF) and its differentiation from psoriasis (PsO) and atopic dermatitis (AD). By utilizing a comprehensive dataset of 1659 skin images, including cases of MF, PsO, AD, and normal skin, a novel multi-frame AI algorithm was used for computer-aided diagnosis. The automatic segmentation and classification of skin lesions were further explored using advanced techniques, such as U-Net Attention models and XGBoost algorithms, transforming images from the color space to the spectral domain. The potential of AI and HSI in dermatological diagnostics was underscored, offering a noninvasive, efficient, and accurate alternative to traditional methods. The findings are particularly crucial for early-stage invasive lesion detection in MF, showcasing the model’s robust performance in segmenting and classifying lesions and its superior predictive accuracy validated through k-fold cross-validation. The model attained its optimal performance with a k-fold cross-validation value of 7, achieving a sensitivity of 90.72%, a specificity of 96.76%, an F1-score of 90.08%, and an ROC-AUC of 0.9351. This study marks a substantial advancement in dermatological diagnostics, thereby contributing significantly to the early and precise identification of skin malignancies and inflammatory conditions.

## 1. Introduction

Cutaneous T-cell lymphomas (CTCLs) are a group of rare dermatological malignancies of idiopathic origin, characterized by the neoplastic proliferation of T lymphocytes that play a vital role in the immune defense against pathogenic microorganisms. In CTCLs, aberrant T-cell growth predisposes the skin’s stratified epithelium to bacterial infiltration. The incipient phase of the disease is known as Mycosis fungoides (MF), which typically manifests as erythematous and scaling plaques, predominantly in sun-protected areas, or as waxing and waning nodules; these lesions disseminate systemically as a result of the migration of malignant T cells, including Sézary cells [1,2,3,4]. Early intervention is paramount to mitigate the symptomatology of the disease and decelerate its evolution. However, the premycotic presentation of MF often mimics the clinical features of psoriasis (PsO) and atopic dermatitis (AD), thereby complicating the differential diagnosis [5,6,7]. Contemporary diagnostic modalities for MF encompass the histopathological examination of cutaneous biopsies [8,9,10,11], peripheral blood analyses [12,13,14,15], and flow cytometric evaluation of T-cell clonality [13,16,17,18], complemented by lymph node biopsies and cross-sectional imaging such as computed tomography (CT) of the thoracoabdominal cavity [19,20] or fluorodeoxyglucose positron emission tomography (FDG-PET) [21,22,23,24]. In current clinical practice, the evaluation of MF predominantly relies on the histopathological analysis of tissue biopsies, lymphatic imaging modalities, hematological assessments, and various other invasive diagnostic techniques. This multifaceted approach to diagnosis, while thorough, is inherently complex and protracted, thereby imposing a considerable psychological burden on the patient.

In contemporary research, the image interpretation capabilities of artificial intelligence (AI) models have been demonstrated to attain a level of diagnostic accuracy that is comparable to that of traditional, invasive testing methodologies, thus, indicating their potential utility as adjunct diagnostic tools. This study explored the utilization of AI for computer-aided diagnosis, employing a singular multi-frame algorithm to discern three distinct dermatological disorders—MF, PsO, and AD—by analyzing cutaneous lesion imagery [25]. Validation against histopathological specimens and optical coherence tomography (OCT) corroborated the AI algorithm’s proficiency, showing a 93% accuracy in the detection of these conditions. The model’s diagnostic sensitivity notably varied across diseases, exhibiting superior performance for MF due to the typically larger lesion size. Meanwhile, the identification of AD was comparatively less precise due to the diminutive and less conspicuous nature of the lesions. The implementation of deep learning and object detection algorithms in dermatological diagnosis has garnered considerable interest within the scientific community. A notable enhancement has been reported in the Single Shot MultiBox Detector (SSD) algorithm by Wang et al. [26], published in 2023. This study introduced the SSD-KD, an innovative technique for classifying skin diseases via knowledge distillation, which capitalizes on intra-instance relational features and self-supervised dual relational knowledge distillation to augment the efficiency of lightweight models operating within computational constraints. Empirical evaluations on the extensive ISIC 2019 skin disease dermoscopic image dataset revealed that the SSD-KD achieved a commendable 85% accuracy in categorizing eight varied skin conditions while optimizing the model’s parameter efficiency and computational demands.

The academic research community has shown a growing interest in the development of methodologies for locating skin lesions and segmenting affected areas, a task of paramount importance in dermatological diagnostics. Sumithra et al. [27] introduced a novel approach for the automatic segmentation and classification of skin lesions. Their method incorporates filtering to eliminate extraneous elements, region-based segmentation, and feature extraction for classification. The technique demonstrated promising results, achieving F-measures of 46.71% using a Support Vector Machine (SVM), 34% with k-Nearest Neighbors (k-NN), and 61% for a fusion of the two, tested on a dataset comprising 726 samples from 141 images across five different disease categories. Pandiyan et al. [28] addressed the crucial challenge of classifying skin conditions, a task with significant health implications if not promptly and accurately addressed. Their methodology employed a dynamic graph cut algorithm for skin lesion segmentation, paired with a Naïve Bayes classifier for disease classification. The proposed approach outperformed existing state-of-the-art methods, achieving accuracy rates of 94.3% for benign cases, 91.2% for melanoma, and 92.9% for keratosis, as evaluated on the ISIC 2017 dataset. This advancement holds considerable potential for aiding medical professionals in the early diagnosis and treatment of various skin conditions, thereby enhancing patient care outcomes. Wen et al. [29] proposed the challenging task of skin lesion segmentation in dermoscopic images, a critical step in the precise analysis of melanoma. This network demonstrated its superiority over existing methods across multiple skin lesion segmentation datasets, marking a substantial advancement in the field. However, traditional convolutional neural networks (CNNs) have shown limitations in capturing global context information, often resulting in suboptimal segmentation outcomes.

However, the reliance on spatial information in image processing methodologies presents several limitations, particularly in the context of dermatological imaging. The diversity in photographic data, stemming from varying angles of capture and the multifarious presentation of skin lesions, poses a significant challenge. As illustrated in Figure 1, lesions associated with MF, PsO, and AD exhibit considerable variability in their anatomical distribution. They can manifest in areas ranging from the upper body with small lesion sizes (approximately 5 cm^2^) to extensive involvement of the back, chest, flanks, or lower body. Moreover, these lesions present a wide array of morphologies, from superficial erythematous streaks to deeper epidermal disruptions, some of which may be complicated by infection and bleeding. Given these factors, the automatic localization of lesions by using spatial information alone is not deemed essential. The static nature of the photographic data further diminishes the necessity for employing object-detection algorithms such as YOLOv5 or CNN-based approaches. Clinicians are typically able to identify the location of lesions with relative ease on the basis of surface examination. Most deep-learning CNN algorithms leverage spatial information for feature recognition. However, this approach may not yield definitive results due to the diverse anatomical distribution and morphological heterogeneity of the lesions. Despite these challenges, distinguishing between these three disease types based solely on skin color during clinical examination can be arduous. Consequently, a spectral discriminant method that capitalizes on color space conversion techniques and utilizes the spectral domain is urgently needed. Color space conversion involves translating a color’s representation from one basis to another to ensure that an image, when converted from one color space to another, retains its original appearance as closely as possible. Such an approach could differentiate the extent of invasive lesions on the basis of skin color variations, thus offering a more discernible advantage. The adoption of advanced methods, such as converting from the color space to the spectral domain, represents a noninvasive diagnostic solution that has the potential to achieve high classification accuracy.

In this study, a transformation algorithm from color space to the spectral domain was employed by utilizing hyperspectral imaging (HSI) algorithms to address the challenging task of detecting early-stage invasive lesions in T-cell skin cancer, namely MF, as opposed to milder skin conditions such as AD and PsO. As depicted in Figure 2, input images were initially subjected to coarse segmentation via a Unet-Attention mechanism. Precise accuracy in segmentation is not paramount; the primary objective is to delineate the affected skin from the healthy skin for subsequent analysis within the HSI model. This approach conserves model capacity and enhances accuracy by eliminating the noise caused by moles and other skin pigments. The output from the HSI model underwent dimensional reduction, and it was then classified using the XGBoost algorithm. This framework offers a sophisticated method for distinguishing three commonly confused skin lesions by converting images to the spectral domain within a compact model while maintaining a high level of classification accuracy.

## 2. Materials and Methods

### 2.1. Sample Preparation

A total of 1659 skin images were collected for training purposes. This dataset included 333 images of MF, 421 images of PsO, 351 images of AD, and 554 images of normal skin. These images were categorized into four distinct groups on the basis of the dermatologists’ tissue analyses, forming separate sets for training. Expert data annotation was carried out in accordance with the evaluations of pathology slices and OCT images. A digital camera (E-M10 Mark III by Olympus) was employed for capturing the skin photographs. The imaging process was directly conducted by a physician, focusing on the patient’s lesion sites. Disease classification was based on the observations by the dermatologists who reviewed the pathological tissue sections and confirmed the presence of symptomatic manifestations in the patients.

To ensure the uniform stability of illumination, it is imperative that the patient’s position, camera, and arrangement of light sources remain constant. This measure helps prevent aberration and chromatic aberration. The specific scanning position was clinically determined, based on the patient’s reports of their condition and the treating physician’s direct visual observation. Once the clinical lesions were located, multispectral analysis was conducted at these sites. Patients were instructed to alter their position for imaging, allowing a comprehensive coverage of the upper body from both front and back, akin to bilateral positions. For smaller affected areas, such as private areas, armpits, or arms, patients were asked to position themselves for close-up imaging. The data distribution is depicted in Figure 3, which illustrates the variety in the distribution of skin lesions and highlights the most frequent locations of these lesions on the body. A commonality among the three disease types was the prevalence of lesions on larger skin areas, such as the chest and back. Following these were lesions commonly found on the lateral aspects of the chest, shoulders, and arms. Some close-up shots were included to enhance the color accuracy of the lesions.

### 2.2. Segmentation Task

U-Nets are a widely utilized model for image segmentation. They are noted for their effectiveness across various medical imaging applications and their efficiency upon deployment. The structure of a U-Net enables high segmentation accuracy even with a limited number of training samples, thus distinguishing it from other models. Its architecture is founded on multi-stage cascaded convolutional neural networks, and these layers are tasked with extracting regions of interest and providing dense predictions. This makes a U-Net particularly adept at delivering superior performance in medical image segmentation. The attention mechanism, commonly employed in natural language processing, utilizes class-specific pooling to enhance the accuracy and speed of classification tasks. These attention maps emphasize pertinent regions, thus making them more significant for specific health-related datasets and improving the model’s applicability and performance in medical contexts.

The U-Net segmentation model was developed using PyTorch and trained using Binary Cross-Entropy (BCE) Loss with Logits. The model underwent training for 200 epochs by utilizing the Adam optimizer. The training parameters included a learning rate of 0.001, a weight decay of 0.0005, and a batch size of 32.

### 2.3. Model HSI-Feature Extraction Task

In this study, HSI was employed using a digital camera (E-M10 Mark III/Olympus) for skin photography and a spectrometer (Ocean Optics, QE65000) to establish the correlation between these two devices by capturing a shared target. The standard 24-color card (X-Rite Classic, 24 Color Checkers) served as the common target. This card encompasses a range of natural colors, representative of various natural objects such as human skin, leaves, and the blue sky. The digital camera captures these color blocks and translates them into digital images, while the spectrometer concurrently measures light from the 24 color blocks to delineate spectral information. The resultant images were then converted into a set of sRGB channel values. A transformation matrix was derived from these datasets. The entire modeling process is depicted in Figure 4.

#### 2.3.1. Spectrometer to *XYZ* Color Space Conversion

In spectrometry, the transformation of spectral data *R(λ),* ranging from 380 nm to 780 nm, into the *XYZ* color space involves the light source spectrum *S(λ)* used in the camera and the *XYZ* color matching functions $\bar{x}(\lambda )$, $\bar{y}(\lambda )$, and $\bar{z}(\lambda )$ (Appendix A). The *Y* component of the *XYZ* space significantly correlates with brightness. The maximum *Y* value from the light source spectrum is scaled to 100 to calibrate for luminance, setting a brightness ratio *k* as detailed in Equation (4). This scaling ensures consistent luminance representation within a standard range. The spectral data are converted into the *XYZ* value (*XYZ_spectrometer_*) and regulated in the *XYZ* color gamut space by using Equations (1)–(3) as follows:(1)$$X=k\int_{380\;nm}^{780\;nm}S\left(\lambda \right)R(\lambda )\bar{x}(\lambda )\;d\lambda ,$$
(2)$$Y=k\int_{380\;nm}^{780\;nm}S\left(\lambda \right)R\left(\lambda \right)\bar{y}\left(\lambda \right)\;d\lambda ,$$
(3)$$Z=k\int_{380\;nm}^{780\;nm}S\left(\lambda \right)R(\lambda )\bar{z}(\lambda )\;d\lambda ,$$
where *S(λ)* is the spectral density of the sample; *R(λ)* is the spectral density of the standard illuminant; and $\bar{x}(\lambda )$, $\bar{y}(\lambda )$, and $\bar{z}(\lambda )$ are the components of color matching functions. The value *k*, which is the brightness ratio, is given by
(4)$$k=100/\int_{380\;nm}^{780\;nm}S\left(\lambda \right)\bar{y}\left(\lambda \right)\;\;d\lambda$$

#### 2.3.2. Non-Linear *XYZ* Correction

The analysis of the camera’s nonlinear response showed that the spectrum analyzer exhibited a linear response. This investigation focused on the brightness values (denoted as *Y* values) for the 19th–24th color patches on a 24-color checker chart representing the gradient of grayscale changes. A linear regression analysis of the *Y* values corresponding to the *XYZ* color space of these patches revealed the camera’s nonlinearity. Figure 5 illustrates that a third-order polynomial best describes the camera’s nonlinear response, as evidenced by the coefficient of determination (*R^2^*) value. The *R^2^* value for the third-order polynomial regression was 0.99997, which was higher than that of linear and quadratic polynomial regressions. Consequently, the camera’s nonlinear response can be effectively corrected through a third-order polynomial model. The correction factor for the nonlinear response is denoted as *V_non-linear_*, as shown in Equation (5).
(5)$$V_{non-linear}={\left[X^{3}{\;Y}^{3}\;Z^{3}\;X^{2}\;Y^{2}\;Z^{2}\;X\;Y\;Z\;1\right]}^{T}$$

Noise in a camera image refers to the combined spatial and temporal fluctuations in the recorded signal, under the assumption of consistent, even lighting. Dark current, which is influenced by both time and temperature, can lead to the accumulation of noise on the camera sensor. Consequently, it was treated as a constant in the analytical model. This constant is denoted as *V_dark_*, as defined in Equation (6) as follows:(6)$$V_{dark}=\left[\alpha \right]$$

Color noise, also known as chrominance noise, represents a random deviation in color relative to the image’s original colors. In contrast to luminance noise, color noise is typically linked to sensor warming or heating. In the analysis of color noise, an issue arises with respect to color matching. Given that the camera image was transformed into the *XYZ* color space, accounting for the interdependencies among the *X, Y*, and *Z* values became imperative. These relationships were encapsulated within the *XYZ* color-matching functions, which are graphically represented in Appendix A. The functions $\bar{x}$, $\bar{y}$, and $\bar{z}$ were integrally linked to the spectral power distribution. Consequently, a comprehensive enumeration of the potential interactions among *X, Y*, and *Z* was conducted, and the collective outcomes were encapsulated within the variable *V_color_*, as delineated in Equation (7).
(7)$$V_{color}={\left[XYZ\;XY\;YZ\;XZ\;X\;Y\;Z\right]}^{T}$$

After a thorough assessment of all identified error sources was conducted, the variable *V_color_* served as the foundational element for the application of non-linear response correction, through multiplication with *V_non-linear_*. The resultant product was then normalized to the third order to preclude the possibility of over-correction. Subsequently, the constant *V_dark_* was incorporated, leading to the formation of the variable matrix *V*. This process is systematically delineated in Equation (8). The variable matrix *V* was then reintroduced to derive the correction matrix *C*, which constituted the final corrective framework for the system.
(8)$$V=[X^{3}{\;Y}^{3}\;Z^{3}\;X^{2}Y{\;X}^{2}Z\;Y^{2}Z\;XY^{2}\;XZ^{2}\;YZ^{2}{XYZ\;X^{2}{\;Y}^{2}\;Z^{2}\;XY\;XZ\;YZ\;X\;Y\;Z\;\alpha ]}^{T}$$

#### 2.3.3. Correction Matrix *C* and the Calibration Camera and Spectrometer

Error factors, such as nonlinear response, dark current, and color noise, may occur when the camera is shooting. Therefore, the correction matrix *C*, which can be used to correct the camera, was finally obtained, as shown in Equation (9):(9)$$\left[C\right]=\left[{XYZ}_{spectrometer}\right]\times pinv\left(\left[V\right]\right),$$
where *V* is defined as Equation (8) with *X*, *Y*, and *Z* components derived from *XYZ_camera_*, and [*XYZ_spectrometer_*] is the matrix created by *X*, *Y*, and *Z* components obtained from the spectrometer.

Equation (8) was employed to extend the *XYZ_camera_* matrix to the *V* matrix, resulting in the corrected values of *X, Y*, and *Z* (denoted as *XYZ_correction_*), as delineated in Equation (10). Given that the wavelength band utilized in this study falls within the visible light spectrum (380–780 nm), the outcome of this correction can be characterized in terms of chromatic aberration.
(10)$$\left[{XYZ}_{correction}\right]=\left[C\right]\times \left[V\right],$$
where *V* is defined as Equation (8) with *X*, *Y*, and *Z* components derived from *XYZ_camera_*, and [*C*] is the correction matrix obtained from Equation (9).

#### 2.3.4. Principal Component Analysis of Reflectance Spectrum

In the process of converting the *XYZ* values (denoted as *XYZ_correction_*), which are acquired subsequent to camera calibration, into spectral data, principal component analysis (PCA) was employed on the reflectance spectrum data (*R_spectrometer_*) of the standard 24-color chart. This analysis yielded the principal components and their associated scores (including eigenvalues) of the reflectance spectrum for the 24-color checker chart.

The results of PCA on the reflectance spectrum (*R_spectrometer_*) demonstrated that the initial 12 principal components (EVs) accounted for 99.99% of the total variance within the data (Figure 6) and their density details in the spectrum dataset (Appendix A). The cumulative explained variance plot displays the total variance captured by successive principal components. When the plot reaches its 12th point at 99.99%, it signifies that the first 12 components collectively explain 99.99% of the dataset’s variance. Dimensionality reduction was achieved by leveraging these 12 principal components, thereby extracting the principal component scores. The subsequent multivariate regression analysis utilized these scores, with *V_color_* serving as a predictor variable. *V_color_* was chosen due to its comprehensive delineation of the correlations among *X, Y,* and *Z* values. This analytical approach facilitated the derivation of the transformation matrix *M*, which bridges the measurement gap between the camera and the spectrometer, as explicated in Equation (11).
(11)$$\left[M\right]=\left[score\right]\times pinv\left(\left[V_{color}\right]\right)$$
where *V_color_* is defined as Equation (7) with *X*, *Y*, and *Z* components derived from *XYZ_spectrometer_*; [*score*] is the principal component scores obtained from 12 sets of principal components for dimensionality reduction of *R_spectrometer_*; and *R_spectrometer_* is the spectral signal obtained from the spectrometer of 24-color patches ranging from 380 nm to 780 nm with a 1 nm resolution. 

#### 2.3.5. The Hyperspectrum

The hyperspectrum is denoted by *S_spectrum_*, as indicated in Equation (12):(12)$$[S_{spectrum}]=\left[EV\right]\left[M\right]\left[V_{color}\right]$$
where *V_color_* is defined as Equation (7) with *X*, *Y*, and *Z* components derived from *XYZ_correction_*; [*EV*] is the 12 principal components obtained from the dimensionality reduction of *R_spectrometer_*; *R_spectrometer_* is the spectral signal obtained from the spectrometer of 24-color patches ranging from 380 nm to 780 nm with a resolution of 1 nm; and [*M*] is the transformation matrix obtained from Equation (11).

### 2.4. Classification Task

By utilizing dimensionality reduction based on the t-Distributed Stochastic Neighbor Embedding (t-SNE) model [30], the feature extraction capabilities of the HSI model were assessed, and an appropriate classification method was determined. As depicted in Figure 7, the t-SNE analysis was conducted using 5000 data point samples from three distinct diseased skin groups. The model exhibited a distinct clustering of features specific to the PsO skin group in comparison to the other groups. However, minimal overlap was observed between the feature clusters of the AD and MF groups. Overall, the t-SNE model demonstrated relatively clear segregation among the three skin disease groups. These results showed that the HSI model possesses the capacity for effective high-level representation classification among various skin disease groups. The visualization outputs of the t-SNE model informed the selection of the XGBoost method as a feature extraction and classification technique.

Extreme Gradient Boosting (XGBoost) is known for its high accuracy in supervised learning problems, often comparable to that of other deep learning models. One of the key strengths of XGBoost lies in its rapid training capabilities, enabled by parallel computation. This efficiency is particularly advantageous when utilizing GPU resources, making the algorithm well-suited for handling high-dimensional data such as hyperspectral data. At its core, XGBoost is built upon the decision tree ensembles algorithm, allowing it to process training data without the need for normalization. This approach is rooted in a strategic focus. Instead of uniformly training on all samples, XGBoost prioritizes subsets that demonstrate poor performance. This selection process is skewed towards choosing samples that were misclassified in previous iterations rather than randomly selecting at an equal probability. While XGBoost is primarily tailored for discrete data types, such as tabular data, and may not be the optimal choice for unstructured data, its speed of training is a notable advantage over Artificial Neural Networks (ANNs). Moreover, ANNs often rely on a large volume of training samples and benefit significantly from transfer learning, which is a strategy that is less feasible with specialized data, such as that pertaining to skin diseases. This limitation makes XGBoost a more practical choice in scenarios where data specificity and training efficiency are paramount.

### 2.5. Training Strategy and Performance Evaluation

Given that the study involved only 34 patients, employing a fixed division of training, validation, and testing in the proportions of 70%, 20%, and 10% proved to be unsuitable for out-of-sample testing. This finding was particularly evident when considering that only three patients were involved in the testing phase, each representing a distinct disease group. Thus, k-fold cross-validation was utilized to address this challenge and prevent data leakage. In k-fold cross-validation, the original dataset is evenly divided into k subsets. Within these subsets, one is designated as the test set, and the remaining k−1 subsets are used as the training set. This process is repeated for each fold, allowing the model’s performance to be evaluated across all folds. Three different values of k were tested to assess the effect of the k-fold division on the training model: 3, 5, and 7. K-fold cross-validation is particularly beneficial for comprehensive and accurate model evaluation, especially when dealing with limited datasets. The training data are randomly divided into k parts (where k is an integer, typically 5 or 10). The model is then trained k times, with each iteration using one part as validation data and the remaining parts as training data. The final evaluation of the model is the average of the results from these k training iterations. This methodology offers a more objective and accurate evaluation, which is crucial for determining the model’s suitability for the current data and the specific problem at hand.

The commonly used metrics for performance evaluation are accuracy, sensitivity, specificity, F1-score, and area under the receiver operator curve (ROC-AUC). The metrics are shown in Appendix A.

## 3. Results and Discussions

### 3.1. Evaluate the Performance of the HSI Model

The simulated spectrum was juxtaposed with the reflection spectrum (denoted as R_spectrometer_) of the 24-color card. The root-mean-square error (RMSE) was computed to quantify the discrepancy between the two spectra. The calculated average RMSE was 0.0525 (Appendix A). Color blocks 13–18 exemplify the variance between the simulated spectrum and the measured spectrum of the 24-color card, as illustrated in Figure 8. These six color blocks were selected for investigation due to their representation of six filters commonly used in chromatic correction. The comparison between the six typical colors demonstrates the correlation between the measured and simulated spectra, indicating that the spectral reproduction algorithm has successfully produced simulation results approximating the measured spectra. Consequently, it can be inferred that the accuracy of the hyperspectral model in converting from real data is notably effective. The simulated spectrum was transformed into the L*a*b* color space for a comparative analysis using the CIEDE 2000 color difference metric (Appendix A). This comparison yielded an average color difference of 0.28, as depicted in Table 1. The color difference of the 24-color blocks was illustrated on the CIE 1931 chromaticity diagram for a 2° standard observer, as shown in Figure 9. In this diagram, a black line connects two central points, with “red” indicating simulated colors and “green” representing measured colors. The minimal discrepancy between the simulated and actual colors demonstrated the effectiveness of the HSI algorithm in calibrating the correlation between the camera and the spectrometer.

### 3.2. Segmentation Model Results

Figure 10 presents the performance of the U-Net Attention models by using example images from various sites affected by the three skin disease groups. The model effectively distinguished between damaged and normal skin in the segmented areas, which is crucial because the objective of segmentation tasks is to isolate damaged skin from other elements such as normal skin, melanin spots, and areas affected by different injuries (such as bleeding or scratches). Given the limited number of samples, which influences the training efficiency of a deep learning model, excessive focus on model regularization through hyperparameter tuning was not a primary concern. However, metrics, such as the cross-entropy, dice coefficient, and intersection over union (IoU), were utilized to ensure the efficacy of the segmentation model. The training outcomes, as indicated by a cross-entropy of 0.3220, a dice coefficient of 0.8447, and an IoU of 0.8521, demonstrated the robust performance of the model. The metric-based evaluations and visual assessments of the test images confirmed the high effectiveness of the U-Net Attention model in this context.

### 3.3. Classification Task Results

In this study, the performance of the model was rigorously evaluated using k-fold cross-validation with distinct k values of 3, 5, and 7. As shown in Table 2, for k = 3, the model achieved a sensitivity of 85.61%, a specificity of 95.25%, an F1-score of 86.26%, and an ROC-AUC of 0.9051. When the k value was increased to 5, a notable improvement was found in all metrics, that is, the sensitivity, specificity, F-1 score, and ROC-AUC increased to 89.20%, 96.35%, 89.19%, and 0.9270, respectively. Further enhancement was observed with k = 7, where the model reached its peak performance, exhibiting a sensitivity of 90.72%, a specificity of 96.76%, an F1-score of 90.08%, and an ROC-AUC of 0.9351. This progressive improvement in the model performance with increasing k values underscored the effectiveness of k-fold cross-validation in refining the model’s predictive accuracy.

Figure 11 presents the confusion matrices for the k-fold cross-validation at k = 3, 5, and 7, providing insights into the model’s classification accuracy across four categories: MF, PsO, AD, and normal skin. For k = 3, the confusion matrix illustrated a balanced performance across all categories, with notable precision in classifying PsO (37 out of 43) and normal (49 out of 54) cases. However, a few instances of MF and AD were misclassified as other categories. When k = 5, the confusion matrix displayed improved accuracy, especially in correctly identifying all PsO cases (35 out of 39), and a significant reduction in misclassification of normal cases. Finally, at k = 7, the confusion matrix further refined the model’s performance, particularly in the precise identification of AD cases (33 out of 33); though, a slight increase in misclassified cases was observed in the normal category. These matrices collectively demonstrated a gradual enhancement of the model’s ability to distinguish between these dermatological conditions, thus reflecting the efficacy of higher k-fold values in achieving more accurate and reliable classification results.

Figure 12 illustrates the ROC curves for the three datasets, (a) k = 3, (b) k = 5, and (c) k = 7, across four classes (0, normal; 1, MF; 2, PsO; and 3, AD). For k = 3, the AUC values for the classes were impressively high, ranging from 0.94 to 0.97, with the highest for class 2 (PsO) at 0.97, indicating a robust classification capability. In the k = 5 dataset, a noticeable uptick in the AUC values was observed, affirming the model’s consistency and enhanced accuracy, particularly for class 2 (PsO) and class 3 (AD), with AUCs of 0.97 and 0.95, respectively. When k = 7, the model maintained a similar level of performance, with AUC values for class 0 (normal) and class 1 (MF) remaining steady at 0.96, whereas class 2 (PsO) reached a peak AUC of 0.98. The slight decrease in class 3’s AUC to 0.94 in k = 7 indicated a nuanced differentiation in the model’s predictive ability across different folds. Overall, these results highlighted the efficacy of the applied model in distinguishing among the four classes, with subtle variations observed across different k-fold settings.

### 3.4. Comparison with Other Existing Studies

In this section, we critically examine the performance of existing models relative to our proposed ensemble model, which integrates three distinct components: U-Net Attention for segmentation, a Hyperspectral model for feature extraction, and XGBoost for classification. The evaluation of the ensemble model was carried out according to the following criteria: segmentation efficacy, as measured by the IoU index; and classification efficiency, gauged by the ROC-AUC index in comparison to existing models with segmentation capabilities. Additionally, given that our model is designed for rapid inference, the impact on prediction speed was also a crucial factor under consideration.

To assess deployment efficiency, we measured the prediction time for each image and computed the average. Table 3 presents the model’s performance evaluation based on model size, IoU segmentation accuracy, AUC-ROC classification accuracy, and average prediction time per image, aiming to compare the practical deployment efficiency against existing models. The model’s prediction speed, denoted by the number of images processed per second, is a critical efficiency metric. Table 3 indicates that the count of non-trainable parameters is largely dictated by the architectural structure of the segmentation module, specifically the U-Net Attention model. The inclusion of the hyperspectral imaging module has resulted in a slight increase in the model size. Our ensemble model’s segmentation efficiency primarily hinges on the U-Net Attention module. While segmentation is crucial, the priority is not heavily placed on the localization of damaged skin areas but rather on differentiating them accurately from other skin lesions. Consequently, our model excelled in classification efficacy, evidenced by a classification accuracy of 0.9351, achieved using a 7-fold training approach.

In terms of prediction speed, our model demonstrated competent performance at an acceptable rate. It is noted that a trade-off exists between accuracy and inference time; a higher accuracy might lead to an increased inference time. These results highlight that our hybrid models, despite longer deployment times, prioritize diagnostic accuracy, a critical aspect in medical systems.

## 4. Conclusions

In this study, a novel approach leveraging HSI algorithms for the early detection of invasive lesions in T-cell skin cancer, particularly MF, was introduced. This method distinctively contrasts with traditional diagnostic processes for milder skin conditions such as AD and PsO. By utilizing a transformation algorithm from the color space to the spectral domain, coarse segmentation was applied via a Unet-Attention mechanism to the input images. This initial segmentation, while not necessitating high precision, effectively distinguished between affected and healthy skin, thereby optimizing the HSI model’s capacity and accuracy by filtering out irrelevant noise such as moles and other skin pigments. Subsequently, the HSI model’s output was subjected to dimensional reduction and classified using the XGBoost algorithm. This framework presents an advanced and compact solution for accurately differentiating among three frequently misdiagnosed skin lesions by transforming images into the spectral domain.

As described in the Introduction, this study was confined to the identification of early-stage skin cancer, specifically MF, which is frequently misdiagnosed as either of two inflammatory diseases, PsO or AD. Consequently, the primary limitation is that the study’s focus is narrow and susceptible to interference from noise when other skin diseases are present which manifest with characteristics dissimilar to the aforementioned three or presenting with lesions and discolorations on the skin that appear nearly identical to those of the three diseases. Furthermore, another significant constraint is the generalizability of the clinical examination applications due to the limited participant pool of 34 individuals, rendering the sample size insufficiently varied to supplant direct physician diagnosis. The diagnostic system developed herein serves only to augment physician diagnosis, offering a reference to aid in formulating subsequent treatment plans. In addition, limited skin phototypes also impact research outcomes. This study predominantly involved Taiwanese patients with skin phototypes ranging from III (light brown) to IV (moderate brown), according to Fitzpatrick’s classification [31]. Consequently, observations on skin areas with phototypes exceeding IV (dark brown), such as tanned skin, may result in chromatic aberration. Therefore, it is anticipated that future studies will incorporate a more diverse array of skin phototypes to broaden the research dataset. Skin phototype variations are a noted limitation in hyperspectral imaging, a spectral domain signal processing technology where uneven lighting significantly impacts spectral results. To mitigate aberrations, patient positioning, camera setup, and lighting must be consistent, with scanning positions clinically determined and adjusted for comprehensive coverage. White balance and chromatic adaptation algorithms, including the Gray World and Bradford methods, are employed to ensure color uniformity across different skin phototypes, effectively handling chromatic aberration and diverse skin conditions.

In an age where advancements in medical technology are paramount, this study echoes the necessity of continual innovation in the realm of medical diagnostics. Similar to the evolving approach to managing diseases such as dementia and retinal disorders, the application of HSI in dermatology indicates a transformative step in medical imaging. By combining the precision of spectral analysis with the prowess of AI algorithms, this research paves the way for more accurate and early detection of skin cancers, thus setting a new standard in dermatological diagnostics and patient care. The success of this study not only reinforces the potential of hyperspectral conversion technology in skin lesion analysis but also inspires further exploration into its applicability across various medical imaging domains.

## Figures and Tables

**Figure 1 cancers-16-00217-f001:**
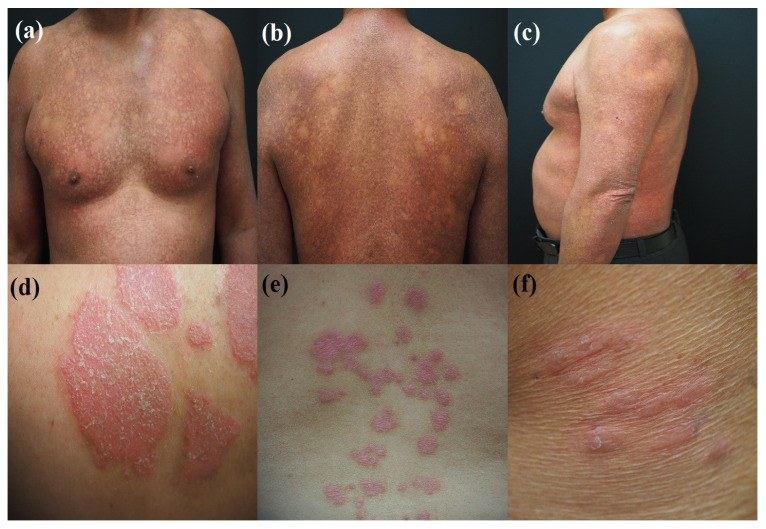
The distribution of skin lesions in MF, PsO, and AD is remarkably varied. These lesions can manifest on the upper body with small areas starting from 5 cm^2^, extending to larger lesions that spread over extensive regions of the (**a**) chest, (**b**) back, (**c**) arms, flanks, or lower body. The morphological presentation of these lesions ranges from superficial red streaks on the skin to deeper lesions in the epidermal layers, which may be associated with infection and bleeding. These lesions, which appear as red clusters, are relatively challenging to distinguish with the naked eye. As depicted in the figure, the lesions are identified as [(**a**–**c**)] MF, [(**d**,**e**)] PsO, and (**f**) AD.

**Figure 2 cancers-16-00217-f002:**
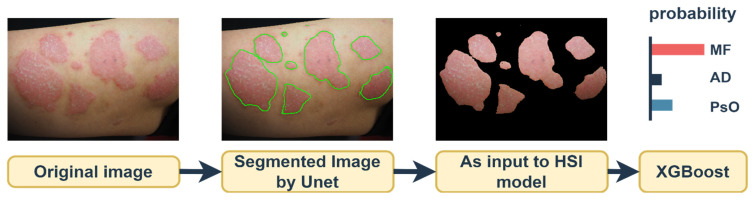
The experimental workflow is as follows: Input images were initially segmented in a coarse manner by using Unet-Attention. The exact precision in this segmentation phase is not critical. The primary aim is to segregate the affected skin from the normal skin for processing in the hyperspectral imaging (HSI) model. This step aids in conserving model capacity and enhances accuracy by filtering out noise, such as that from moles and other skin pigments, which are removed during this stage. Subsequently, the output from the HSI model underwent dimensional reduction, and it was classified using the XGBoost algorithm.

**Figure 3 cancers-16-00217-f003:**
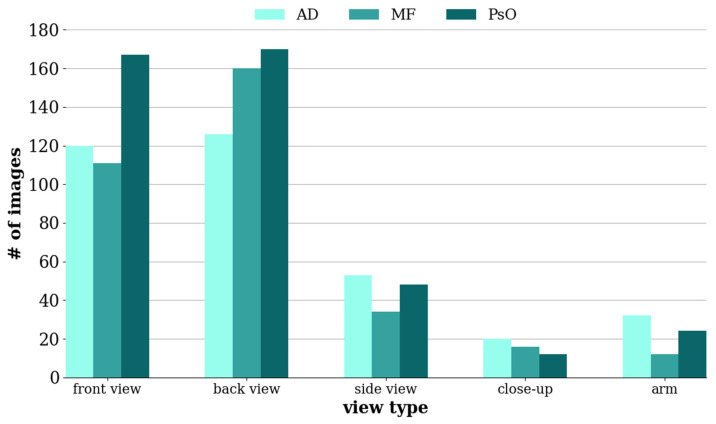
The data distribution displays the number of images in accordance with their shooting angles, encompassing back-view shots, side-view shots, and close-up shots.

**Figure 4 cancers-16-00217-f004:**
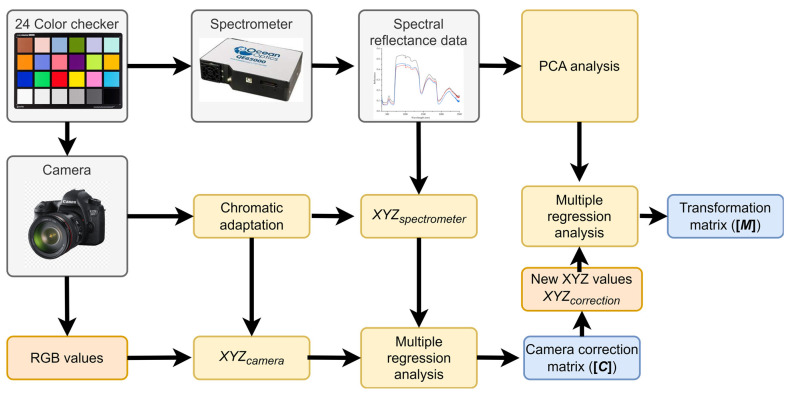
A schematic of the proposed method is displayed. Hyperspectral imaging facilitates the conversion between the camera and the spectrometer, employing standard 24-color blocks (X-Rite Classic, 24 Color Checkers) as calibration references. The digital camera captures these color blocks, translating the information into digital images. Concurrently, the spectrometer delineates spectral information by measuring the light from the 24 color blocks. Subsequently, the camera’s images are algorithmically transformed into a spectrum equivalent to that derived from the spectrometer, utilizing hyperspectral imaging techniques.

**Figure 5 cancers-16-00217-f005:**
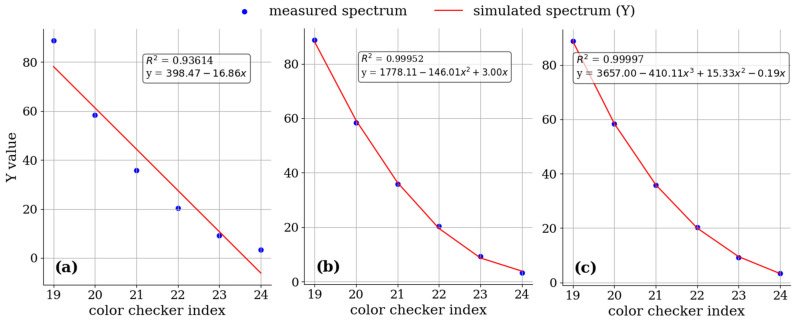
Polynomial regression analysis of the brightness values of these color checker blocks was conducted to represent the gradient of grayscale changes. This analysis comprises (**a**) first-order (linear) regression, (**b**) second-order (quadratic) regression, and (**c**) third-order (cubic) polynomial regression.

**Figure 6 cancers-16-00217-f006:**
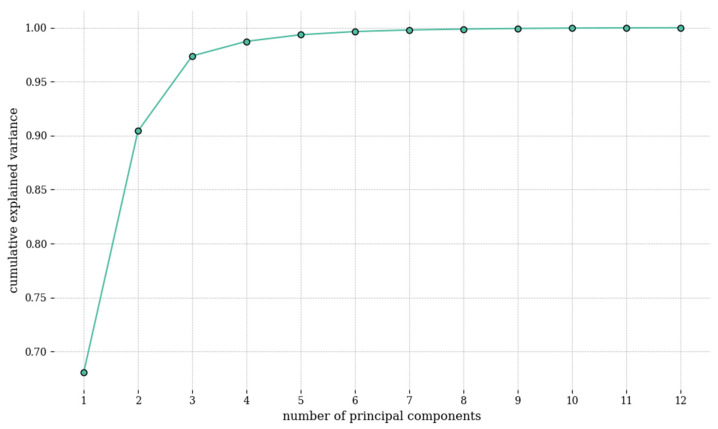
Cumulative explained variance of the first 12 principal components in the spectrum obtained from the principal component analysis.

**Figure 7 cancers-16-00217-f007:**
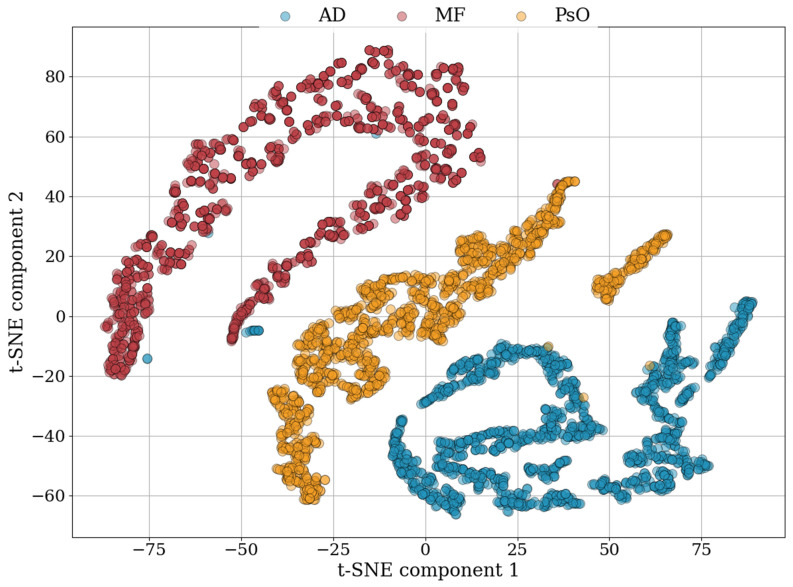
The t-SNE analysis utilized 5000 data points, sampled from three different skin disease groups. This analysis revealed a clear clustering of features unique to the PsO skin group, setting it apart from the other groups.

**Figure 8 cancers-16-00217-f008:**
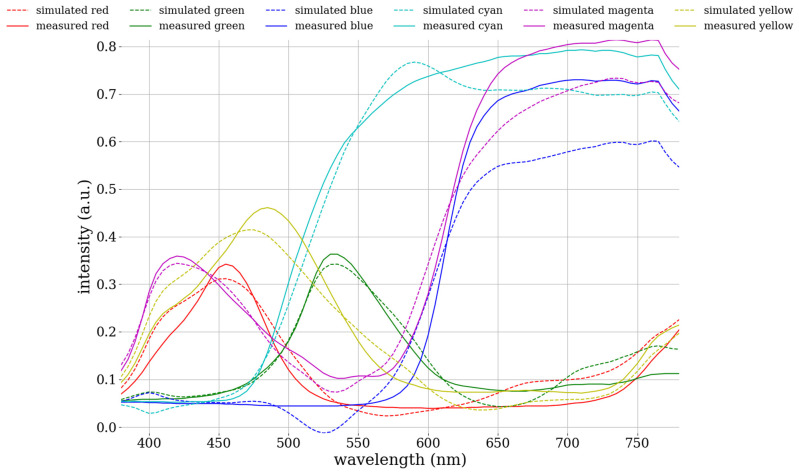
The correlation between the measured spectrum and the spectrum simulated by the algorithm, representing six basic color filters commonly used in chromatic correction, was assessed. This comparison revealed that the algorithm’s simulated reproduction closely matched the measured spectrum. The largest deviation occurred at longer wavelengths, specifically those exceeding 600 nm.

**Figure 9 cancers-16-00217-f009:**
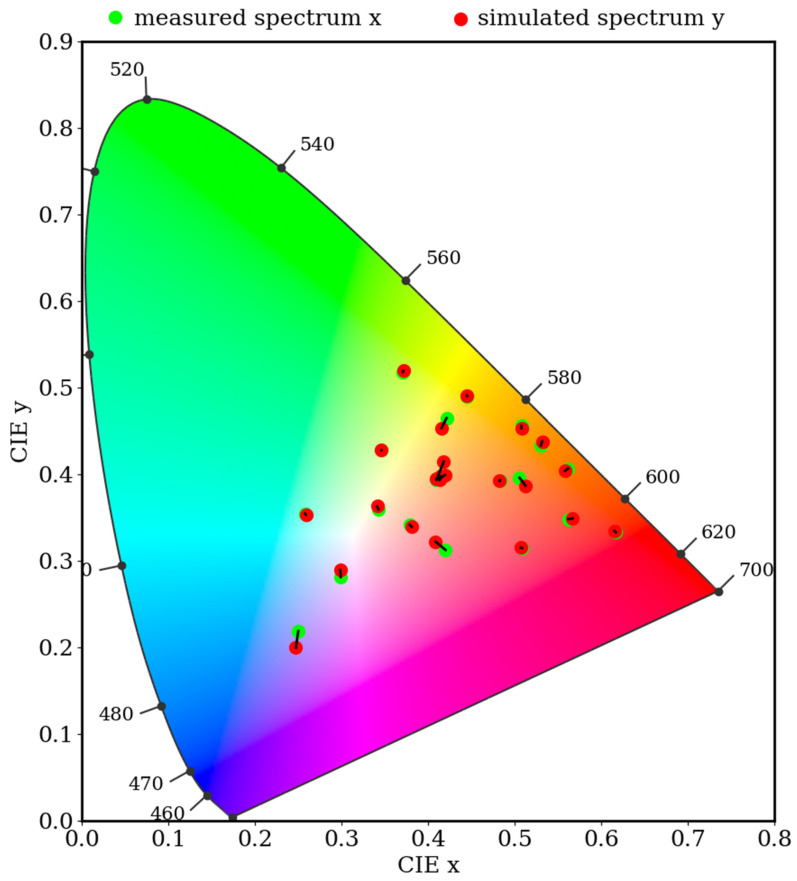
The color difference of 24-color blocks is shown on the CIE 1931 chromaticity diagram 2° standard observer. The black line connecting the center of the two points is “red” to indicate simulated colors and “green” to indicate measured colors.

**Figure 10 cancers-16-00217-f010:**
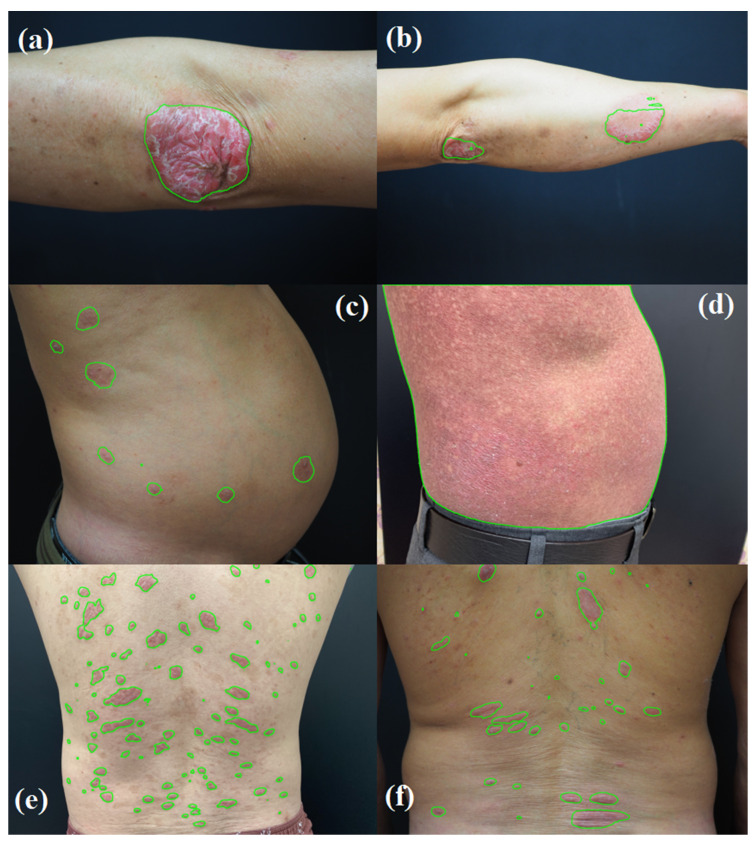
Visualization of the segmentation by the U-Net Attention model illustrates that most lesion areas are effectively isolated from the surrounding normal skin areas. The lesions manifest in various forms: either as close-up images on the arm (**a**,**b**), as clusters comprising numerous small, scattered spots (**c**,**e**,**f**), or even as a large area (**d**) distinctly segmented areas, in contrast to other skin regions.

**Figure 11 cancers-16-00217-f011:**
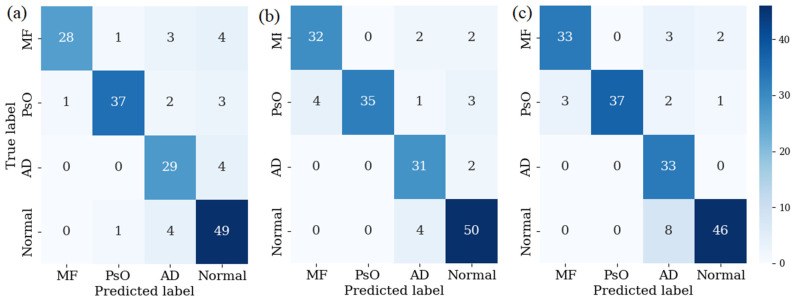
Confusion matrix for k-folds (**a**) 3, (**b**) 5, and (**c**) 7.

**Figure 12 cancers-16-00217-f012:**
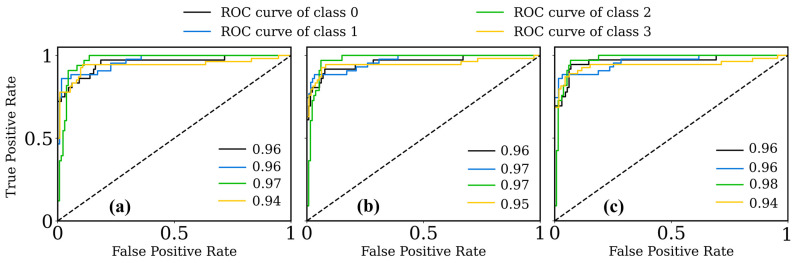
Receiver operating characteristics for (**a**) k = 3; (**b**) k = 5, and (**c**) k = 7 folds.

**Table 1 cancers-16-00217-t001:** Chromatic difference diagram of measured and simulated spectra of 24-color blocks.

Color No.	Measured	Simulated	L* (Measured)	a* (Measured)	b* (Measured)	L* (Simulated)	a* (Simulated)	b* (Simulated)	CIEDE 2000
1			38.94	24.35	36.89	38.81	24.92	38.23	0.52
2			67.93	32.09	49.08	67.95	32.03	48.94	0.05
3			49.45	0.07	9.87	49.34	−0.66	9.68	1.01
4			42.33	−4.05	39.45	42.59	−3.97	38.03	0.57
5			55.89	16.17	11.32	55.88	16.12	11.33	0.04
6			69.10	−19.39	31.48	69.10	−19.46	31.45	0.05
7			65.32	46.33	82.11	65.37	46.43	81.81	0.14
8			39.09	8.89	−10.49	39.23	9.36	−10.40	0.49
9			55.11	58.96	46.87	55.06	59.03	47.19	0.14
10			31.22	24.81	6.34	31.78	25.36	6.05	0.57
11			71.87	−5.83	76.56	71.87	−5.85	76.51	0.02
12			74.87	34.26	90.13	74.84	34.16	90.25	0.08
13			27.87	11.77	−23.10	27.60	11.34	−23.26	0.46
14			53.97	−27.31	50.50	53.95	−27.22	50.74	0.11
15			45.76	66.20	51.52	45.76	65.96	51.01	0.17
16			83.76	23.22	100.15	83.76	23.26	100.18	0.02
17			54.10	58.28	24.01	54.11	58.30	24.01	0.01
18			48.03	−23.70	−0.08	48.08	−23.35	−0.02	0.19
19			95.47	15.52	46.28	95.48	15.52	46.28	0.00
20			80.98	13.58	40.13	80.95	13.62	40.11	0.04
21			66.39	11.35	34.00	66.52	11.47	34.18	0.14
22			52.19	9.40	28.14	51.90	9.29	28.26	0.31
23			36.45	6.84	21.33	36.46	6.79	21.44	0.09
24			21.36	4.87	14.76	21.05	3.68	15.29	1.55
								average	0.28

**Table 2 cancers-16-00217-t002:** Model training performance was evaluated across different k-fold values (3, 5, and 7) with respect to four performance metrics.

k-Fold	Sensitivity	Specificity	F1-Score	ROC-AUC
k3	85.61%	95.25%	86.26%	0.9051
k5	89.20%	96.35%	89.19%	0.9270
k7	90.72%	96.76%	90.08%	0.9351

**Table 3 cancers-16-00217-t003:** Comparison of model parameters and performances.

Model	Number of Params	Size (MB)	IoU	AUC-ROC	Time Prediction per Image (second)
U-Net	31M	124	0.8003	0.8490	0.0741
U-Net++	55M	220	0.8977	0.9334	0.1182
U-Net Attention	35M	140	0.8521	0.9097	0.0792
DeepLabv3	41M	164	0.8730	0.9597	0.0836
Ours (3-folds)	35M	148	0.8521	0.9051	0.0810
Ours (5-folds)	35M	148	0.8521	0.9270	0.0810
Ours (7-folds)	35M	148	0.8521	0.9351	0.0810

## Data Availability

The data presented in this study are available in this article upon request to the corresponding author.

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
