# Peer review of "Identification of Skin Lesions by Snapshot Hyperspectral Imaging"

_cancers, 2024, doi:10.3390/cancers16010217_

Round 1

Reviewer 1 Report

Comments and Suggestions for Authors

Dear authors, congratulation for your idea and for your study. Indeed this is pioneer work and a much needed one. As you may already know, mycosis fungoides has three different stages. The earliest one, plaque mycosis (sometimes called plaque parapsoriasis) can last for years and it is, for definitive reason, called "great imitator". In this stage even skin skin biopsy is not helpful. Early diagnostic is mandatory for influencing the course of the disease and new technologies that can do that are extremely important and much needed.

I have only a few comments/suggestions for you:

- try to discuss the limitations of your study (for sure there has to be some)

- try to discuss how your research can be adapted into clinical use (a device, a method, etc) because your paper is very technical and it is not very clear in this respect

- most important: discuss what are your method's limitations regarding skin phototypes. The same erythema in Type II Fitzpatrick differs (being more bright) than in Type IV or V, etc. Colors differ even in the same patient, on lower limbs erythematous lesions have a bluish nuance (are more dark) because of the stasis.

Reviewer 2 Report

Comments and Suggestions for Authors

The authors reported the results of a study with the aim of pioneering the application of artificial intelligence (AI) and hyperspectral imaging (HSI) in the diagnosis of skin cancer lesions, particularly focusing on Mycosis fungoides (MF) and its differentiation from psoriasis (PsO) and atopic dermatitis (AD). The manuscript is interesting ad well written. The topic is original and relevant in its field. Discussion is supported by the results. Strengths and limitations of the study have been discussed. References are appropriate. Tables and figures improve the quality of the paper. The manuscript is suitable for publication.

Reviewer 3 Report

Comments and Suggestions for Authors

The article “: Identification of skin lesions by snapshot hyperspectral imaging Special Issue: Advances in Oncological Imaging”, deals with the application of artificial intelligence and hyperspectral imaging in the discrimination of skin cancer lesions, in particular Mycosis fungoides from non-cancer lesions. The study is promising to improve the optically based diagnostic procedures in dermatology. However there some faults, as specified below, that make the paper to require a revision.

In general, in the work is used a relatively large number of skin images of diagnosed skin lesions, for the training of the procedure to apply and evaluate then its performance on skin lesions from 34 patients.

The organization of some parts of the paper is to be revised, including the balance between the text on the training and performance check. Also, the text of “Materials and Methods” contains a too much extensive description of the results obtained along the set-up of training procedures, which seem to be more pertinent to a “Results” section. A section about the measured and simulated results and their comparison, and procedure validation could be a first part of the “Results”.  

Figure 1 and related text should be moved to “Results” section, since it represents the interrogation starting point of the work in terms of  variability in anatomical distribution of the skin lesions. Figure 2, in turn, could be relevant to Materials and Methods, “2.2. Segmentation task” section.

Figure 3 legend and related text. In the legend the locations of the lesion sites from which the pictures are taken are to be better described (front or back view of chest? and so on). Also, please explain better “.. shooting angles”.

Figure 4 legend is to be revised, for a clearer description of the “work” performed by the camera and that performed by the spectrograph.

Figure 7 is not present in the text. Also, Figure 7 legend reports on “….representing six basic color blocks typical of common camera filters…”, while along the text it is to describe and specify  about  the use of such filters.

Minor remarks

- Line 151 - Please give the definition of “color space” at first time it is mentioned

- Line 230 – please remove “to”

- Line 230-231 “ …digital camera and the spectrometer were utilized to measure the spectral data of the 24-color card and capture the images.. “ should be :” digital camera and the spectrometer to were utilized to capture the images and measure the spectral data ….”, is it?

Figure 12 – quality and readability are to be improved.

Comments on the Quality of English Language

good

Round 2

Reviewer 3 Report

Comments and Suggestions for Authors

Convincing answers have been provided to the comments of the Reviewer, and where the case the paper has been duly revise. The paper is now suitable for publication in Cancer.

Comments on the Quality of English Language

Good, some sentences to be checked